# An Evaluation of the Lineage of *Brucella* Isolates in Turkey by a Whole-Genome Single-Nucleotide Polymorphism Analysis

**DOI:** 10.3390/vetsci11070316

**Published:** 2024-07-14

**Authors:** Kadir Akar, Katharina Holzer, Ludwig E. Hoelzle, Gülseren Yıldız Öz, Shaimaa Abdelmegid, Emin Ayhan Baklan, Buket Eroğlu, Eray Atıl, Shawky A. Moustafa, Gamal Wareth, Manar Elkhayat

**Affiliations:** 1Faculty of Veterinary Medicine, Van Yuzuncu Yıl University, 65090 Van, Turkey; 2Department of Livestock Infectiology and Environmental Hygiene, Institute of Animal Science, University of Hohenheim, 70599 Stuttgart, Germany; ludwig.hoelzle@uni-hohenheim.de; 3NRL for Brucellosis, Pendik Veterinary Control Institute, 34890 Istanbul, Turkey; gulseren.yildiz@tarimorman.gov.tr (G.Y.Ö.); eminayhan.baklan@tarimorman.gov.tr (E.A.B.); buket.eroglu@tarimorman.gov.tr (B.E.); eray.atil@tarimorman.gov.tr (E.A.); 4Faculty of Veterinary Medicine, Assiut University, Assiut 71515, Egypt; shaimaawagdey@gmail.com; 5Faculty of Veterinary Medicine, Benha University, Toukh 13736, Egypt; dr.shawky.gesriha@gmail.com (S.A.M.); manar.elkhayat@fvtm.bu.edu (M.E.); 6Institute of Bacterial Infections and Zoonoses, Friedrich-Loeffler-Institute, 07743 Jena, Germany; gamal.wareth@fli.de; 7Institute of Infectious Diseases and Infection Control, Jena University Hospital, Am Klinikum 1, 07747 Jena, Germany

**Keywords:** brucellosis, whole-genome single-nucleotide polymorphism analysis, Turkey, in silico analysis

## Abstract

**Simple Summary:**

Brucellosis is a disease that is commonly found in Turkey among both humans and animals. It causes significant economic losses in the livestock industry and is of great public health concern. This study focused on analyzing the genetic makeup of *Brucella* isolates from Turkey using a core-genome single-nucleotide polymorphism (cgSNP) analysis. This analysis showed potential links between the Turkish isolates and those from neighboring countries and other parts of the world. This highlights the importance of implementing strict measures to control the spread of brucellosis throughout the country.

**Abstract:**

Brucellosis is a disease caused by the *Brucella* (*B.*) species. It is a zoonotic disease that affects farm animals and causes economic losses in many countries worldwide. *Brucella* has the ability to persist in the environment and infect the host at low doses. Thus, it is more important to trace brucellosis outbreaks, identify their sources of infection, and interrupt their transmission. Some countries already have initial data, but most of these data are based on a Multiple-Locus Variable-Number Tandem-Repeat Analysis (MLVA), which is completely unsuitable for studying the *Brucella* genome. Since brucellosis is an endemic disease in Turkey, this study aimed to examine the genome of Turkish *Brucella* isolates collected between 2018 and 2020, except for one isolate, which was from 2012. A total of 28 strains of *B. melitensis* (*n* = 15) and *B. abortus* (*n* = 13) were analyzed using a core-genome single-nucleotide polymorphism (cgSNP) analysis. A potential connection between the Turkish isolates and entries from Sweden, Israel, Syria, Austria, and India for *B. melitensis* was detected. For *B. abortus*, there may be potential associations with entries from China. This explains the tight ties found between *Brucella* strains from neighboring countries and isolates from Turkey. Therefore, it is recommended that strict measures be taken and the possible effects of uncontrolled animal introduction are emphasized.

## 1. Introduction

Brucellosis is a bacterial disease that can affect both animals and humans and is able to cause serious problems for public health. This disease can lead to economic losses due to decreased productivity in livestock connected with livestock abortions [1]. The genus *Brucella* encompasses 13 facultative intracellular bacterial species, and all of them are officially recognized by the World Organization for Animal Health (OIE) [2,3]. The two species that occur most frequently are *B. melitensis* and *B. abortus*. While *B. melitensis* primarily affects small ruminants, *B. abortus* is the primary cause of brucellosis in bovines. However, documented cases confirm the occurrence of inter-species transmission [4]. In the Mediterranean basin, Middle Eastern countries, and Turkey, *B. melitensis* has been identified as the dominant species causing brucellosis in both humans and animals [5,6]. Transmission to humans can occur through contact with infected animals, slaughterhouse waste, or the consumption of contaminated food items such as unpasteurized milk and dairy products [7]. The standard treatment for brucellosis involves a prolonged six-week regimen with two different antibiotics, commonly rifampicin and doxycycline, as recommended by the World Health Organization [8]. Despite efforts to control the disease through vaccination programs initiated in 2012 [9], brucellosis persists in Turkey, highlighting the need for vigilant monitoring and the control of *Brucella* species circulation, especially within farm settings. Understanding the genetic relationships of isolates is crucial for identifying infection sources. Proper brucellosis control requires surveillance and highly discriminatory methods to determine the source of infection and spread of outbreak strains since, in Turkey, there is a lack of data about the spreading of *Brucella*.

In this context, we undertook a comprehensive investigation of the genetic relationships among 28 *Brucella* isolates from Turkey, comprising 15 *B. melitensis* and 13 *B. abortus* isolates, to investigate its diversity and distribution. This study utilized the most reliable method currently available, employing a single-nucleotide polymorphism (SNP) analysis based on whole-genome sequencing. Notably, there is a dearth of reliable whole-genome data from Turkey, with only a limited dataset based on MLVA available, a method proven to be outdated and completely unreliable for *Brucella* epidemiology since it was shown that using MLVA leads to error-prone results when applied to the *Brucella* genome. This is because MLVA uses microsatellites, which are prone to homoplasy. That means that the same genotypes can arise through independent mutations in different populations, leading to convergence. Additionally, changes in tandem repeats can occur through subcultures in the laboratory [10,11]. It has been demonstrated that the MLVA can significantly deviate from the cgSNP analysis due to these factors [12]. Furthermore, MLVA has been used for genotyping but cannot accurately trace the origin and transmission of *Brucella* strains [13].

## 2. Materials and Methods

### 2.1. Origin of Bacterial Isolates

Bacterial isolates from animal abortion cases in Turkey were collected during 2018–2022 from Mersin, Erzrum, Kars, Istanbul, Aksaray, Elazig, Sanlrurfa, Izmir, Bayburt, Edirne, Kirklareli, Samsun, Konya, Erzincan, Bursa, Kocaeli, Bilecik, and Adana. One isolate originates from 2012. The isolates were isolated from cattle (*n* = 16), sheep (*n* = 7), goats (*n* = 1), buffaloes (*n* = 3), and one originated from a human. All were isolated from abortion materials (lung, liver, kidney, vaginal swab, and milk). Unfortunately, the organs can no longer be assigned to the specific isolates. All 28 isolates used (15 *B. melitensis* and 13 *B. abortus*) are listed in Table 1. All metadata of the downloaded entries from the Sequence Read Archive (SRA) are listed in Appendix A.

### 2.2. Brucella Isolation from Sample Materials and DNA Extraction

Swab samples were streaked on *Brucella* selective agar and subcultured three times to achieve pure *Brucella* colonies. Study isolates and media panels containing lysis with phages, dyes, and analin were tested for growth properties in the antibiotic. Growth characteristics were analyzed in a medium containing H_2_S production and susceptibility to *Brucella* phage 104 RTD (Tbilisi, R/C, Izatnagar), thionin (20 μg/mL), basic fuchsin (20 μg/mL), penicillin (5 IU/mL), i-erythritol (1 mg/mL), and streptomycin (2.5 μg/mL) [14]. Suspected *Brucella* colonies from the selective agar medium were selected and cultured in a *Brucella* broth for approximately three days for DNA extraction. The DNA extraction process followed the instructions outlined in the commercial kit protocol (High Pure FFPET DNA Isolation Kit, Roche Diagnostics, Mannheim, Germany).

### 2.3. Whole-Genome Sequencing and Bioinformatics Procedure of the Raw Reads

The total genomic DNA sequencing and genomic library preparation were performed by the BMLabosis BM Lab. Sist. Ltd., Şti. (Ankara, Turkey). Data were sequenced using Illumina Novaseq 6000. The Linux-based bioinformatics WGSBAC (v.2.1) pipeline (https://gitlab.com/FLI_Bioinfo/WGSBAC/-/tree/version2, accessed on 22 February 2022) was used to analyze WGS data automatically and as a standard. The pipeline input consisted of the Illumina paired-ended FASTQ files and the metadata file. This was optimized by combining contigs using Shovill v.1.0.4 (https://github.com/tseemann/shovill, accessed on 24 February 2022) for the SPAdes combiner [15]. Quality controls were performed using the combined contigs QUAST v. 5.0.2 [16]. Genomic features are shown in Appendix A. All FASTQ files were submitted to the National Center for Biotechnology Information (NCBI) under project number PRJNA1071326. For a comparison with isolates from other countries, public entries from the Sequence Read Archive (SRA) (https://www.ncbi.nlm.nih.gov/sra, accessed on 8 February 2024) were downloaded. These downloads consisted of raw reads from *B. melitensis* or *B. abortus* originating from multiple countries. These sequences were also subjected to the same pipeline and were included in the tree construction.

### 2.4. Species and Antimicrobial Resistance Determination

The *Brucella* isolates from different hosts were determined by the Bruce-ladder PCR [17,18,19] in silico to define their species via Geneious v.11.1.5 (https://www.geneious.com/, accessed on 16 May 2022) with contigs assembled using the Shovill program in the bioinformatics pipeline.

The in silico detection of antimicrobial resistance genes was performed on the genome assemblies generated by the pipeline, utilizing the NCBI AMR Finder Plus Database (https://github.com/ncbi/amr/wiki/Running-AMRFinderPlus, accessed on 16 May 2022) [20] to identify resistance genes and chromosomal mutations mediating antimicrobial resistance. 

### 2.5. Canonical SNP Assay

To determine the African, American, East Mediterranean, or a not-yet-defined clade (probably the West Mediterranean [WM] clade, according to personal communication with Jeffrey T. Foster, developer of the canonical SNP [canSNP] assay, University of Arizona) of *B. melitensis*, the canSNP assay was performed according to Foster et al. [21] in silico using the Geneious v.11.1.5. Primer, which was matched to the complementary genome sequence using the contigs provided by the bioinformatics pipeline described in Section 2.3. As described by Foster et al. [21], the GC overhang for the primers is unnecessary for the in silico analysis and was, therefore, omitted. After primer annealing, the respective nucleotide position of the isolate can be determined. This assay was developed for *B. melitensis*, but for *B. abortus*, unfortunately, there is no equivalent assay.

### 2.6. Core-Genome SNP Genotyping

In silico, a single-nucleotide polymorphism (SNP) search was performed using Snippy (v. 4.6.0) with default parameters (https://github.com/tseemann/snippy, accessed on 22 February 2022) for the Turkish isolates. Detection of SNPs was performed on the core genome without rRNA genes to exclude highly mutable regions. The core-genome SNP (cgSNPs) (Appendix A) was called based on alignment to the 2308 reference strain for *B. abortus* and the 16M reference strain for *B. melitensis* (GenBank accession numbers NC_007618.1, NC_007624.1, NC_003317.1 and NC_003318.1). Core-genome tSNP-based genotypes (cgSNPGTs) were determined using a maximum of one cgSNP difference. In addition, in the same procedure, the Turkish isolates were compared this time with the data downloaded from the SRA. For *B. melitensis*, there were 705 downloaded entries from the SRA, originating from the USA, Israel, Italy, Austria, Egypt, Serbia, India, Kuwait, Syria, Iraq, Iran, Afghanistan, Morocco, Romania, Algeria, Bulgaria, Albania, Turkmenistan, Jordan, Saudi Arabia, Somalia, China, and Sweden. For *B. abortus*, there were 363 downloaded entries from the USA, Egypt, India, China, the UK, and Bangladesh.

The creation of minimum spanning trees (MSTs) was performed with Bionumerics version 8.0 (Applied Maths, Schaerbeek, Belgium) based on character data. The trees were permutated 1000 times and rooted by the maximum branch length. The MSTs are presented with logarithmic scaling. Bionumerics uses Prim’s algorithm for establishing MSTs.

## 3. Results

### 3.1. Origin of B. melitensis and B. abortus Isolates

Fifteen of the 28 *Brucella* isolates in total examined in this study were confirmed to be *B. melitensis*, and 13 were confirmed to be *B. abortus* (Table 1) by the in silico Bruce-ladder PCR. All isolates originate from animals except one. Among the isolates, the isolate Bru9 was classified as the S19 vaccine strain. Figure 1 shows the strains’ geographic distributions.

### 3.2. Antimicrobial Resistance

The in silico analysis showed that Bru17 (*B. melitensis*) presents a *rph*C of the *rph* gene family, encoding for rifampicin phosphotransferases, indicating a probable rifampicin inactivating mechanism and a *mer*R1 gene, encoding a mercuric resistance operon regulatory protein. Bru22 (*B. abortus*) encodes a beta-lactamase (*bla*TEm181), thereby expressing resistance to antibiotics containing a beta-lactam ring.

### 3.3. Canonical SNP Analysis

The Turkish isolates corresponded mostly to the East Mediterranean clade except the isolates Bru10, 16, 17, and 18, which belong to the American clade, according to Foster et al. [21]. The clade for Bru12 could not be determined. Bru10, 16, 17, and 18 differ from Bru12 by just one cgSNP and from the next isolates from the East Mediterranean clade (Bru25) by only four cgSNPs. The nearest isolates (Bru1) differ from the East Mediterranean clade (Bru10, 16, 17, and 18) by 20 cgSNPs and Bru3 by 17 cgSNPs.

### 3.4. Core-Genome SNP Genotyping for B. melitensis

The aforementioned 15 *B. melitensis* isolates from Turkey were compared to 700 downloaded entries from the SRA, originating from 24 countries from the Middle East, Europe, and the USA (Figure 2). Notably, a distinct subdivision into two clusters is observed, separated by 358 core-genome single-nucleotide polymorphisms (cgSNPs). The left cluster primarily includes, among others, the Italian isolates, representing Mediterranean strains that intermingle with Egyptian strains. A close association between Egyptian *Brucella* isolates and those from Italy has previously been demonstrated [12]. Additionally, this cluster also includes isolates from Austria, two from Sweden, and one from Algeria, the USA, and Morocco.

The cluster on the right, distinguished by 358 cgSNPs, comprises isolates from Turkey, Israel, Serbia, India, Romania, Kuwait, Syria, Iraq, Iran, Algeria, Bulgaria, Jordan, Serbia, Albania, Saudi Arabia, Turkmenistan, Afghanistan, Somalia, and also Sweden and Morocco. Two more isolates from the USA are found in the lower cluster, one corresponding to the reference genome 16M.

The left cluster exhibits a maximum cgSNP distance of 29 within itself, while the cluster on the right has a cgSNP distance of 61. The reference strain (from the USA) is located at the bottom of the right cluster, each with a cgSNP distance of 47 compared to one isolate from Turkey, one isolate from Austria, and one isolate also from the USA.

### 3.5. Core-Genome SNP Genotyping for B. abortus

The 13 aforementioned *B. abortus* isolates were compared to the 363 downloaded entries (raw reads) from the SRA (Figure 3). In this case, as well, it is evident that the isolates are separated into two clusters, distinguished by 242 core-genome single-nucleotide polymorphisms (cgSNPs). The first major cluster primarily comprises isolates from the USA, UK, and Egypt, with two individual isolates, each from India and Turkey. The other smaller cluster consists of isolates from Turkey, each with one isolate from China and Bangladesh. The solitary isolate from Bangladesh also exhibits a significant cgSNP distance of 172 cgSNPs when compared to the Turkish isolates, in addition to the remaining cgSNP distances.

Interestingly, the reference strain is identical to an isolate from Turkey (Bru5, isolated from cattle in 2019).

## 4. Discussion

Brucellosis is an important zoonotic disease that is endemic in the Mediterranean basin, with a high prevalence in Turkey [5,22]. Brucellosis is a global One Health problem that can affect international trade, allowing it to spread from endemic areas to other parts of the world [23]. In this study, the distribution and possible spread of *B. abortus* and *B. melitensis* epidemic isolates were analyzed at the genome level. As a result, the cgSNP analysis revealed a high resolution among the data, allowing for an examination of the outbreak strains and offering a perspective on the distribution and potential spread of epidemic isolates. With 13 *B. abortus* and 15 *B. melitensis* isolates collected from diverse locations over four years, our study presents a contribution to understanding the genomic landscape of *Brucella* in Turkey. The current comprehensive investigation provides insights into the genetic diversity and relationships among Turkish *Brucella* isolates since former studies based on genotyping are very limited in this country [5,24]. This study is among the few WGS-based studies conducted in Turkey and is important because it underscores the necessity of performing more WGS-based analyses. Increasing the number of records analyzed in the database and conducting detailed genetic analyses of isolates circulating in the region will enable a deeper interpretation of the area’s genetic landscape.

### 4.1. Comparison of B. melitensis Outbreak Strains to Strains from Other Countries

Some studies report that *B. melitensis* is a highly prevalent and expanding latent “traveling bacterium” [25,26]. In Turkey, *B. melitensis* has been reported to be the most severe strain causing brucellosis [27]. In Figure 2, two clusters can be seen, differentiated by 358 cgSNPs. This means that the isolates of both clusters may not share a very close genetic relatedness, indicating that the infection dynamics should probably be evaluated separately. The cluster of isolates from Turkey exhibits small cgSNP differences. The isolates closely related to those from Turkey are from Sweden, Iraq, Syria, Kuwait, Austria, Serbia, Israel, India, and Afghanistan. There is also one isolate from Albania, Romania, the USA, Egypt, Iran, and Saudi Arabia. Reference strain 16M (from the USA) is located at the bottom of the right cluster compared to a total of three isolates from Turkey, Austria, and the USA. Therefore, these countries may share a possible common source of infection, suggesting a transmission route across borders. Unfortunately, there is a lack of publicly available official data for tracking or cross-referencing, such as animal transport between these countries or transport for slaughter processes, to support this theory. The only source that provides information on animal imports into Turkey is the study by Yücer et al. [28]. It demonstrates that cattle and heifers were imported into Turkey from Uruguay, Brazil, Hungary, the Czech Republic, France, Australia, and other unspecified countries from 2010 to 2019. Unfortunately, there are no relevant *B. melitensis* isolates in public databases to compare with the Turkish isolates and to determine if there were possible contaminations between the countries. However, Iran, Iraq, Syria, and Bulgaria are direct neighboring countries of Turkey, reinforcing the hypothesis of spatial transmission, which is compatible with the study involving isolates from Iraq by Massis, which defined the Iraqi strains as belonging to the East Mediterranean clade [29]. The present study revealed that most of the collected Turkish isolates belong to the East Mediterranean clade, according to the assay by Foster et al. [21], which aligns with the study from the neighboring country Iraq [29]. However, our study also identified isolates from the American clade, suggesting that Turkey may have had two distinct original sources of infection. The close proximity of isolates from the East Mediterranean clade and the American clade is surprising. Bru10, 16, 17, and 18 (American clade) and Bru25 (East Mediterranean clade) differ by only 4 cgSNPs. One possible explanation is that during genome comparison in the program Snippy, only the single nucleotides at positions present in all genomes are considered, significantly shrinking the genome and reducing the cgSNP distances. However, a distance of just 4 cgSNPs is still remarkably small and was unexpected.

The fact that the territory used to belong to the Ottoman Empire, that there was intensive livestock trade with the southern borders before the Gulf War, and that border controls were weaker before Turkey’s attempt to join the EU may all be related [30,31]. In a study conducted in Sweden, despite the country being officially free from brucellosis, some reported samples were examined, and it was found that most of the patients had a history of traveling to Syria and Iraq. These cases were frequently associated with Turkish genotypes, which is consistent with our study [32]. Notably, China, India, Somalia, Turkmenistan, and Saudi Arabia, which do not share borders with Turkey and are geographically distant, also have isolates potentially circulating with the same outbreak strain. The historical existence of brucellosis, revealed by metagenomic studies, shows that it is compatible with the profiles of isolates from Central Asian and Far Eastern countries, as well as the Eastern Mediterranean countries and Turkey [33]. One study’s findings draw attention to the fact that countries along the Silk Road may have the same or similar epidemiological features of brucellosis [26]. Further investigation is needed to understand why these countries may share a common source of infection.

Upon closer examination, there is even an isolate from Turkey (Bru12, isolated from a sheep in 2019) that shares the same genotype as an isolate from Austria and the USA. Additionally, isolates Bru10, Bru16, Bru17, and Bru18 from Turkey share the same genotype as an isolate from Italy. A reason for sharing the same genotype may be the fact that Turkish citizens travel abroad or live in Italy since it is known that Turkey does not export animals or animal products to European countries. This theory could be supported by the fact that the *B. melitensis* isolate Bru17 (goat, 2020) presents a rifampicin-inactivating phosphotransferase and a mercuric resistance operon regulatory protein, indicating a treatment history. Similarly, two other isolates from Turkey, downloaded from the database, share the same genotype as isolates from Syria and Sweden and one isolate from Iraq. A highly probable transmission or spread between those countries must have occurred, and they may share a common source of infection. It is likely that Turkey, which has a high *Brucella* incidence and borders Syria, had an illegal entry of many refugees and animals into the country during the war in Syria in 2011, or this may be due to the frequent livestock trade with both Iraq and Syria before the Gulf War [31,34,35].

According to the MST in general, it can be inferred that Austria must have had two different sources of infection, as isolates from this country are represented in both clusters. In an Austrian study, it was reported that the Austrian isolates belong to the Western Mediterranean lineage. In the same study, it was reported that one isolate was transmitted from cheese produced from raw milk in Turkey, which was associated with a Turkish isolate. Therefore, its presence in that region could be attributed to the large number of travelers and refugees [34]. This could explain the presence of the Austrian isolate in each cluster. Similarly, Sweden, which has two isolates in the left cluster along with isolates from Italy and Egypt and one isolate from the USA, also suggests the presence of dual sources of infection since both isolates from the USA have a difference of 47 cgSNPs. The strong trade link between Italy and North Africa, dating back to the Roman Empire, has been explained as the history of the introduction of brucellosis between Africa and the Northern Mediterranean region [36].

Notably, there are two isolates from Italy and one from Egypt in the right cluster. It is likely that *Brucella* strains from the countries mentioned in the right cluster were introduced to Italy and Egypt. Furthermore, Austria and Italy exhibit a very close genetic relatedness in terms of genotypes, possibly due to their neighboring status, indicating that both countries are possibly infected with the same outbreak strain due to spatial proximity. The isolates from Egypt also show a potential close relatedness to those from Italy and Austria. *Brucella* isolates circulating in Italy originate from various lineages, predominantly from the Western Mediterranean lineage, but it has been reported to originate from the Eastern Mediterranean lineage centuries ago [36]. Isolates belonging to the Eastern Mediterranean lineage were also detected. Metagenomic analyses of the trade relationship between Italy and Egypt also confirm the proximity of the isolates between the regions [36]. Another study reveal that the same causative species was previously reported to spread between Italy and Egypt [12]. This publication highlighted the transportation of cattle and buffaloes for breeding from Italy to Egypt between 1986 and 2018, suggesting a possible source of infection. Morocco, Algeria, and Sweden may possibly share a common outbreak strain due to close proximity in the left cluster.

### 4.2. Comparison of B. abortus Outbreak Strains to Strains from Other Countries

*B. abortus* is, as well as *B. melitensis*, the most common *Brucella* species in domestic animals [26]. Nonetheless, human-induced brucellosis cases in Turkey are mainly caused by *B. melitensis* [37]. As is the case for *B. melitensis,* in the current study, *B. abortus* isolates were also divided into two clusters when compared to isolates from other countries. The first large cluster, which includes all the isolates from the USA, UK, Egypt, and one isolate from India, exhibits a clear demarcation with at least 242 cgSNPs from the remaining isolates from Turkey, as well as both individual isolates from China and Bangladesh and the reference strain 2308 from the USA. The high cgSNP count of 242, compared to the other cgSNP distances within the single- to double-digit range within this cluster, likely suggests that the outbreak in Turkey may not be closely related to the isolates from the USA, UK, Egypt, and India. The Turkish outbreak is less likely to share a close kinship with isolates from these mentioned countries except for one exception: one isolate from Turkey is indeed present in the large cluster as well (Bru9, isolated from cattle in 2019). This suggests that Turkey may have two sources of *B. abortus* infection: one from the USA, which has not yet spread widely, or the USA introduced the infection from Turkey, and this strain has since proliferated extensively in the USA. The first scenario seems more probable.

Interestingly, a Turkish isolate (Bru11, isolated from a buffalo in 2019) exhibits exactly the same genotype as the reference strain 2308 from the USA. Moreover, the Turkish isolates could be related to the isolate from China, differing by 41–46 cgSNPs from the Turkish isolates. It could be that a delay or mix-up has occurred here, either from China or the other way around. The standalone isolate from Bangladesh, differing by 172 cgSNPs from the Turkish isolates, could already be considered a separate outbreak. Furthermore, the *B. abortus* isolate Bru22 possesses a beta-lactamase activity, indicating that the hosts most probably were treated with the mentioned antibiotics.

It is regrettable that there is no animal import and export data available for Turkey from these countries, which would allow us to trace the history and epidemiology, making these infections unfortunately untraceable. Also, some previous studies have demonstrated the necessity of using a whole-genome SNP analysis to identify differentiated outbreak strains and imported *B. abortus* strains [38,39,40]. However, the isolates in our study from the USA, UK, China, Bangladesh, and India date back to earlier times (1985–2017) than the *Brucella* isolates from Turkey (2018–2022). In a study conducted in China for countries on the Silk Road, the genotypic similarities of *B. abortus* strains with isolates from China and Kazakhstan, Mongolia, Italy, and Turkey were emphasized [26]. These data may explain the compatibility of our study samples with Chinese isolates. Nevertheless, this does not provide a definitive conclusion, as the time of isolation does not necessarily correspond to the time of infection or introduction. Thus, the USA, UK, Egypt, and India may potentially share a common source of infection, as these isolates within the large cluster exhibit relatively short cgSNP distances (one- or two-digit range) from each other.

In summary, there may be three different sources of infection in this context: one circulating in Bangladesh, one in Turkey and China, and one in the UK, the USA, Egypt, and India.

### 4.3. Outlook

Brucellosis remains prevalent in Middle Eastern countries, and several factors contribute to the failure of eradication efforts [41]. In the future, it would be especially interesting if directly neighboring countries such as Iran, Iraq, Syria, Greece, Georgia, and Armenia conducted more detailed examinations of *Brucella* outbreak strains. Currently, there are sparse data for *B. melitensis* and no data for *B. abortus*, making it difficult to track outbreaks accurately. The comprehensive documentation of outbreaks, as well as records of the import and export of animals for breeding and slaughter purposes, could be of great importance. Additionally, the global documentation of these events would be very helpful. A significantly larger amount of genetic data is necessary worldwide to understand the relatedness of individual strains, such as determining how closely the Chinese and Turkish strains, which differ by 41 cgSNPs, are related to each other in order to differentiate outbreak strains more precisely.

## 5. Conclusions

Brucellosis is an endemic disease in Turkey, as well as in other Mediterranean countries. This study was conducted because there were no or scarce data on the SNP basis for Turkish isolates so far. In this study, the genomes of self-isolated as well as downloaded genomic data from the SRA database of *B. melitensis* and *B. abortus* were compared using the cgSNP analysis. By comparing our panel of isolates with entries from public databases, the results show possible relationships, especially with entries from Sweden, Israel, Syria, Austria, and India for *B. melitensis* and possibly with entries from China for *B. abortus*. The endemic nature of the disease in Turkey highlights the necessity for the implementation of rigorous measures in areas where there is close interaction between humans and animals carrying the disease agent. This is attributed to the close relationship observed between the *Brucella* isolates from Turkey and the strains from the neighboring nations. Consequently, this implies the importance of taking preventive measures and underscores the potential significance of uncontrolled animal introductions as well.

## Figures and Tables

**Figure 1 vetsci-11-00316-f001:**
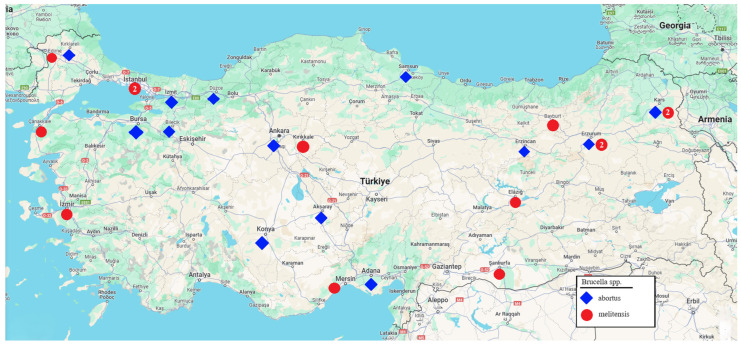
Geographical distribution of the isolates. Red circles indicate the cities where the *B. melitensis* isolates and the blue rhombuses indicate the cities where *B. abortus* isolates were taken. No number indicates that only one isolate was taken, while the number two indicates that two isolates were taken.

**Figure 2 vetsci-11-00316-f002:**
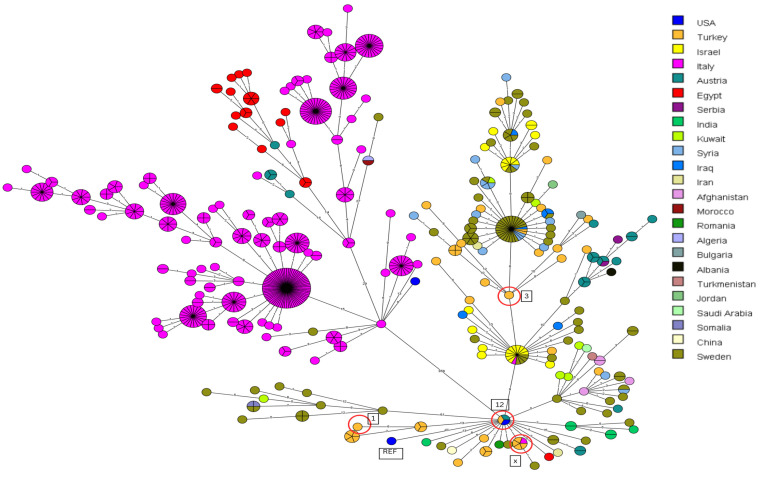
MST based on cgSNP analysis generated from the 15 Turkish *B. melitensis* isolates used in this study and the 700 available entries from the SRA database. *B. melitensis* 16M was set as a reference genome (REF). The same color represents the same country. The isolates in orange are the ones from Turkey. The numbers on the lines represent the cgSNP distances. The different clades are illustrated in this image. The number 1 represents the isolate Bru1 (East Mediterranean clade); the number 12 represents Bru12; the letter X represents the isolates Bru10, 16, 17, and 18 (American clade); and the number 3 represents Bru3 (East Mediterranean clade).

**Figure 3 vetsci-11-00316-f003:**
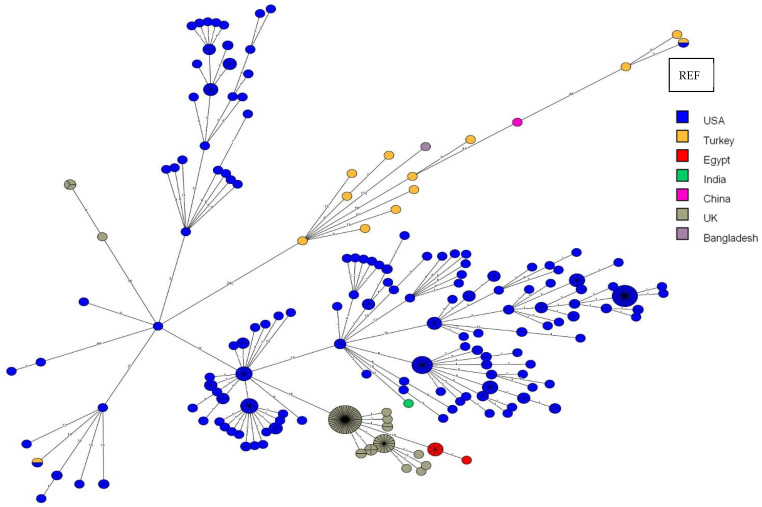
MST based on cgSNP analysis generated from the 13 Turkish *B. abortus* isolates used in this study and the 363 available entries from the SRA database. *B. abortus* 2308 was set as the reference genome (REF). The same color represents the same country. The isolates in orange are the ones from Turkey. The numbers on the lines represent the cgSNP distances.

**Table 1 vetsci-11-00316-t001:** Metadata of *B. melitensis* and *B. abortus* isolates from Turkey was analyzed in the current study.

Strain ID	Species	Source of Isolation	Host	Geographical Area	Year of Isolation
bru1	*B. melitensis*	Organ	Cattle	Mersin	2021
bru2	*B. melitensis*	Organ	Cattle	Erzurum	2018
bru3	*B. melitensis*	Organ	Cattle	Kars	2012
bru4	*B. melitensis*	Organ	Sheep	Istanbul	2019
bru8	*B. melitensis*	Organ	Cattle	Elazığ	2019
bru10	*B. melitensis*	Organ	Cattle	Şanlıurfa	2019
bru12	*B. melitensis*	Organ	Sheep	Izmir	2019
bru14	*B. melitensis*	Organ	Cattle	Kars	2020
bru16	*B. melitensis*	Organ	Cattle	Bayburt	2020
bru17	*B. melitensis*	Organ	Goat	unknown	2020
bru18	*B. melitensis*	Organ	Sheep	Edirne	2021
bru20	*B. melitensis*	Organ	Cattle	Erzurum	2021
bru23	*B. melitensis*	Organ	Human	Istanbul	2022
bru25	*B. melitensis*	Organ	Sheep	Kırıkkale	2022
bru27	*B. melitensis*	Organ	Sheep	Canakkale	2022
bru5	*B. abortus*	Organ	Cattle	Aksaray	2019
bru6	*B. abortus*	Organ	Cattle	Ankara	2019
bru7	*B. abortus*	Organ	Buffalo	Düzce	2019
bru9	*B. abortus*	Milk	Cattle	Kırklareli	2019
bru11	*B. abortus*	Organ	Buffalo	Samsun	2019
bru13	*B. abortus*	Organ	Sheep	Konya	2019
bru15	*B. abortus*	Organ	Cattle	Erzincan	2020
bru19	*B. abortus*	Organ	Cattle	Bursa	2021
bru21	*B. abortus*	Organ	Sheep	Erzurum	2018
bru22	*B. abortus*	Organ	Buffalo	Kocaeli	2018
bru24	*B. abortus*	Organ	Cattle	Bilecik	2022
bru26	*B. abortus*	Organ	Cattle	Adana	2022
bru28	*B. abortus*	Organ	Cattle	Kars	2022

## Data Availability

All FASTQ files were submitted to the National Center for Biotechnology Information (NCBI) under project number PRJNA1071326.

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
