# Peer review of "An Evaluation of the Lineage of Brucella Isolates in Turkey by a Whole-Genome Single-Nucleotide Polymorphism Analysis"

_vetsci, 2024, doi:10.3390/vetsci11070316_

Round 1

Reviewer 1 Report

Comments and Suggestions for Authors

Comments on the Quality of English Language

Author Response

Dear Reviewer,

We want to thank you very much for the thorough corrections and your suggestions, which will undoubtedly improve our manuscript. We really greatly appreciate your feedback. Please find below our responses to your comments. We corrected the text thorough and hope that it will satisfy you.

Answers to Reviewer 1:

  1. The abstract should briefly mention the specific outcomes of the SNP analysis for clarity.

    Answer: Thank you very much for this information. The abstract is written like this:

    „A potential connection between the Turkish isolates and entries from Sweden, Israel, Syria, Austria, and India for B. melitensis were detected. For B. abortus, there may be potential associations with entries from China. This explains the tight ties found between Brucella strains from neighboring countries and isolates from Turkey.“

The outcome has already been mentioned. If you wish to make other changes, please specify exactly what is meant.

  1. More recent studies on brucellosis should be included to reinforce the relevance of the research.

    Answer: We included more recent studies and corrected the whole discussion part. We also included one more analysis; canonical SNP analysis.

  1. Instead of merely citing the previous work, it would be better to do detailed comparison with previous research to highlight study contribution and differences.

Answer: It is certainly a good approach. However, unfortunately, there are no comprehensive studies on the situation in Turkey based on SNP analysis, making a thorough comparison difficult. Additionally, this publication, through the SNP analysis, is intended to serve merely as an initial reference point. However, as mentioned before, we added some recent studies and corrected the manuscript.

  1. Türkiye" should be consistent with either "Turkey" or "Türkiye" throughout the document.

    Answer: We corrected this.

  1. Line 34: "A total of 28 strains of Brucella melitensis (n=15) and Brucella abortus (n=13) were analyzed...". Ensure consistency with species names. Italicize them: Brucella melitensis, Brucella abortus.

Answer: We made the corrections.

  1. Line 60- Highlighting should be Highlighted.

Answer: In our opinion, and after review by a professional English speaker, we believe the word „highlighting“ is correct.

  1. Line 83 - “All of these isolates were isolated from abortion materials”- should be “All were isolated from abortion material”

Answer: We corrected this.

  1. Line 62. “ is” should be replaced with “are”.

Answer: We corrected this.

  1. "In silico analysis shows that Bru 17 (B. melitensis) presents a rphC gene..." Clarify gene function and significance."

Answer: We adapted the text according to your suggestion.

  1. Line 336.- “Exhibts should be exhibit”.

Answer: We corrected this.

  1. Provide statistical method for data analysis in result section.

Answer: Since we excluded our mistake that in the manuscript was written, that we performed some phylogenetics, we excluded this mistake from the text. We just performed Minimum Spanning Trees (MSTs). We wrote that the creation of the MSTs was performed with Bionumerics version 8.0 (Applied Maths, Belgium) based on character data. The trees were permutated 1000 times and rooted by maximum branch length. The MSTs are presented with logarithmic scaling. We additionally added that Bionumerics uses Prim’s algorithm for establishing MSTs. We hope this satisfy you.

  1. Suggest specific areas for future research based on the study finding.

    Answer: Thank you for this suggestion. We added an outlook part in the manuscript and mentioned specific areas.

  1. We underwent a professional correction in the English language.

Reviewer 2 Report

Comments and Suggestions for Authors

MANUSCRIPT VETERINARY SCIENCES 3060245 PEER REVIEW

The study examined the genomes of Turkish Brucella isolates from 2018-2020, plus one from 2012. Using core genome single nucleotide polymorphism (cgSNP) analysis, 28 strains were analyzed: 15 Brucella melitensis and 13 Brucella abortus. Connections were found between Turkish B. melitensis isolates and those from Sweden, Israel, Syria, Austria, and India, and between Turkish B. abortus and strains from China. This suggests close ties between Brucella strains from neighboring countries and Turkey. Consequently, strict measures are recommended to control the introduction of animals to prevent the spread of brucellosis.

The manuscript is well written, and the research looks sound. However, I think the manuscript requires revision before final acceptance. This mostly come from the fact that there is plenty of sentences and ideas supported by no evidence, clear examples, or appropriate references. Without proper revision, I will consider the study rather speculative. In this sense, I have a few comments I hope the authors may consider, as follows:

INTRODUCTION

Lines 70-73. The authors claim that, from Türkiye, there is a lack of reliable Brucella whole genome data, with “only a limited dataset based on Multi Locus Variable Number of Tandem Repeats (MLVA) available”. The authors also claim that the method is outdated and unreliable for Brucella epidemiology, then they refer to a couple of citations, with no further explanation whatsoever. I suggest the authors to provide with a synthetic (very short, one or two sentences) explanation to backup such statements. Also, please elaborate on why SNPs are a better option.

MATERIALS AND METHODS

Line 85. Please define SRA at first mention.

Table 1. Bellow the “Geographical area” column, are all those Turkish cities? Why do no refer to them that way?

Line 141. There are no genomic features shown in Table 1. Please refer to the appropriate table.

Lines 174-177. There is no single explanation on the way these analyses were performed. Please elaborate on each of them (specific methods used, parameters used, parameters values, etc. etc.). How is it supposed these results can be reproduced?

RESULTS

Lines 184-185. This was already mentioned in the Materials and Methods section. There is no need to mention it back again in here.

Figure 1. It must be improved. It needs a better resolution, scale, north-south axis, etc. Avoid unnecessary names and overlapping red/blue squares over names. What about samples outside Türkiye?

Figure 2. Names within figure are too small. I can zoom in to read them but that should not be necessary. Some green and magenta colors are too similar. I cannot tell the difference between Italy and China, for example. Figure 3 has similar issues.

Where are the cluster analyses, phylogenetic analyses, etc. etc. the authors claimed they performed, as mentioned earlier in the Materials and Methods section? Please clarify. Only the minimum spanning trees are presented and described.

DISCUSSION

Lines 246-247. This is the third time the authors mention this. It is redundant and unnecessary.

Lines 248-250. The authors refer to “a detailed examination of the phylogenetic structure” and to a “unique perspective on the distribution and potential spread of epidemic isolates”; however, it is not clear to what analyses are they referring. Therefore, I suggest the authors to elaborate on this.

Lines 251-254 The authors describe their study as a “significant contribution” (line 252), “valuable insights” (lines 253-254), and “important” (line 256). However, there is no descriptions that allow the reader to understand the specific reasons of such statements. I suggest the authors to clearly refer to specific results, their interpretations, and how such results allow to answer specific questions.

Lines 273-275. As the authors pointed out, this has been already mentioned. There is no discussion of the results in here. I suggest the authors to get rid of redundant sentences like this throughout the whole text.

Lines 278-288. Of course, all isolates are related. They all belong to the same species after all. They are also related spatially, no matter if such isolates are coming from neighboring countries or from abroad. There is no need of any analyses to make such statements. What would be interesting is to describe specific relationships. In this sense, I think that the MST the authors present may not be enough to this end. Perhaps if they show their phylogenies, the relationship among isolates (in terms of basal vs tip OTUs or ancestral/derived OTUs), would be more evident. The tone of the remaining paragraphs (until section 4.3) seems to me rather speculative. Therefore, I suggest the authors to reconsider other alternatives (such the already mentioned phylogenies) to present and interpret their results.

Section 4.3. Comparison of B. abortus outbreak strains to strains from other countries. The tone of this section is that of a description, but not a direct discussion of their results. I understand that there may not be enough information on animal importation and exportation, but I think the authors may benefit from other published studies that surely deal with similar questions.

Author Response

Dear Reviewer,

We want to thank you very much for the thorough corrections and your suggestions, which will undoubtedly improve our manuscript. We really greatly appreciate your feedback. Please find below our responses to your comments. We corrected the text thorough and hope that it will satisfy you.

Answers to Reviewer 2:

  1. Introduction: Lines 70-73. The authors claim that, from Türkiye, there is a lack of reliable Brucella whole genome data, with “only a limited dataset based on Multi Locus Variable Number of Tandem Repeats (MLVA) available”. The authors also claim that the method is outdated and unreliable for Brucella epidemiology, then they refer to a couple of citations, with no further explanation whatsoever. I suggest the authors to provide with a synthetic (very short, one or two sentences) explanation to backup such statements. Also, please elaborate on why SNPs are a better option.

    Answer: We added the explanation.

  1. Materials and methods:
  • Line 85. Please define SRA at first mention.

Answer: We followed your suggestion.

  • Table 1. Bellow the “Geographical area” column, are all those Turkish cities? Why do no refer to them that way?

Answer: Yes these are all the official names of the Turkish cities.

  • Line 141. There are no genomic features shown in Table 1. Please refer to the appropriate table.

Answer: Thank you very much for this information. We corrected the text and referred to table S1 in 2.3.

  • Lines 174-177. There is no single explanation on the way these analyses were performed. Please elaborate on each of them (specific methods used, parameters used, parameters values, etc. etc.). How is it supposed these results can be reproduced?

    Answer: There is, of course, a mistake. We deleted the mentioned cluster analysis and phylogeny. We only performed Minimum Spanning Trees, which are described in the following sentences.

  1. Results:

  • Lines 184-185. This was already mentioned in the Materials and Methods section. There is no need to mention it back again in here.

    Answer: We deleted it.

  • Figure 1. It must be improved. It needs a better resolution, scale, north-south axis, etc. Avoid unnecessary names and overlapping red/blue squares over names. What about samples outside Türkiye?

Answer:  We replaced the map. Since we are focusing on isolates from Turkey in our manuscript, and we isolated them ourselves, we only included the map of the isolation from Turkey. The comparison with downloaded isolates from other countries comes from public databases. These were not collected by us and therefore were not included in the map.

  • Figure 2. Names within figure are too small. I can zoom in to read them but that should not be necessary. Some green and magenta colors are too similar. I cannot tell the difference between Italy and China, for example. Figure 3 has similar issues.

    Answer: We replaced similar colors. Due to the large amount of data, the image naturally becomes smaller and less clear. Unfortunately, there is nothing we can do about this. We have now enlarged it by presenting it in landscape format. Additionally, the image will be submitted digitally, allowing for zooming in and viewing everything in sharp resolution.

  • Where are the cluster analyses, phylogenetic analyses, etc. etc. the authors claimed they performed, as mentioned earlier in the Materials and Methods section? Please clarify. Only the minimum spanning trees are presented and described.

    Answer: Yes, you are completely right. As mentioned before, we delete the mentioned cluster analyses and phylogenetic trees in the material and methods part.

  1. Discussion:

  • Lines 246-247. This is the third time the authors mention this. It is redundant and unnecessary.

    Answer: We deleted this.

  • Lines 248-250. The authors refer to “a detailed examination of the phylogenetic structure” and to a “unique perspective on the distribution and potential spread of epidemic isolates”; however, it is not clear to what analyses are they referring. Therefore, I suggest the authors to elaborate on this.

    Answer: We deleted these words and adpated the text. The whole discussion part was correcetd and improved.

  • Lines 251-254 The authors describe their study as a “significant contribution” (line 252), “valuable insights” (lines 253-254), and “important” (line 256). However, there is no descriptions that allow the reader to understand the specific reasons of such statements. I suggest the authors to clearly refer to specific results, their interpretations, and how such results allow to answer specific questions.

    Answer: We adapted the text following your advice.

  • Lines 273-275. As the authors pointed out, this has been already mentioned. There is no discussion of the results in here. I suggest the authors to get rid of redundant sentences like this throughout the whole text.

Answer: We corrected the text.

  • Lines 278-288. Of course, all isolates are related. They all belong to the same species after all. They are also related spatially, no matter if such isolates are coming from neighboring countries or from abroad. There is no need of any analyses to make such statements. What would be interesting is to describe specific relationships. In this sense, I think that the MST the authors present may not be enough to this end. Perhaps if they show their phylogenies, the relationship among isolates (in terms of basal vs tip OTUs or ancestral/derived OTUs), would be more evident. The tone of the remaining paragraphs (until section 4.3) seems to me rather speculative. Therefore, I suggest the authors to reconsider other alternatives (such the already mentioned phylogenies) to present and interpret their results.

    Answer: As mentioned before it was a mistake that we wrote that we did phylogenetic analysis. We did not made this. This manuscript should be a small overview and a first comparison with other countries based on SNP analysis. Therefore, the best solution is to establish a MST. However, we adapted the text according to your suggestions.

  • Section 4.3. Comparison of B. abortus outbreak strains to strains from other countries. The tone of this section is that of a description, but not a direct discussion of their results. I understand that there may not be enough information on animal importation and exportation, but I think the authors may benefit from other published studies that surely deal with similar questions.

    Answer: We adpated the text a little bit. Unfortunatelly, there are no studies based on SNP analysis from Turkey. This make things hard to compare. This manuscript is thought to make a short overview of the circulating isolates from Turkey in comparison with isolates from other countries. However, we included more recent studies, compared our results and improved the whole discussion part.

Round 2

Reviewer 2 Report

Comments and Suggestions for Authors

I think tha authors fullfill mi suggeestions.

Author Response

Dear Academic Editor,

We want to thank you very much for the thorough corrections and your suggestions, which will undoubtedly improve our manuscript. We really greatly appreciate your feedback. Please find below our responses to your comments. We corrected the text thorough and hope that it will satisfy you.

Answers to Editor:

1) Line 33-34 and 73-74: The way MLVA is spelled out is different these two places. In fact, I believe that multiple locus variable number tandem repeat analysis is the correct spelling, although other spellings are also found in the literature, such as variable-number with a hyphen or tandem-repeats with a hyphen or repeats (plural) instead of repeat. Anyway, at least be consistent.

Answer: L. 33-34: was changed to Multilocus Variable Number Tandem Repeat Analysis (MLVA) and L.73-74 was changed to MLVA.

2) Line 76: Brucella in italics

Answer: Edited in italic.

3) L. 82: delete Garofalo, it is sufficient with the reference number.

Answer: Garofalo deleted.

4) L. 170, 173, 200: in silico should be in italics.

Answer: Edited in italic.

5) Results: results should be presented in past tense. You have used present tense on several occasions. Please modify.
Answer: Minor modifications have been made in the results section.

6) L. 209: in silico in italics.
Answer: Edited in italic.

7) L. 209-213: This is a long sentence which is difficult to read. Please rephrase.
Answer: The sentence was split into two sentences.

8) L. 227: 227: What do you mean by "get mixed up with"? Maybe something like: These were interspersed with the other......
Answer: This sentence was deleted.

9) References: The authors should go through all references and adjust to the journal style. E.g., All words in titles should not be in italics (e.g. reference 2, 20, 23).
Answer: All references used were arranged according to the journal rules.

10) Species names should be in italics (applies to several references).
Answer: All species names in the references are italicized.

11) Reference 3: The first ) should be a (
Answer: Corrected.

12) Reference 39: journal name should be abbreviated.

Answer: Journal name abbreviated.